# Iterative VAE as a predictive brain model for out-of-distribution generalization

**Victor Boutin**[1,2], **Aimen Zerroug**[1,2], **Minju Jung**[2], **Thomas Serre**[1,2]

[1] Artificial and Natural Intelligence Toulouse Institute, Université de Toulouse, France
[2] Carney Institute for Brain Science, Dpt. of Cognitive Linguistic & Psychological Sciences
Brown University, Providence, RI 02912
{victor_boutin, aimen_zerroug, minju_jung, thomas_serre}@brown.edu

## Abstract

Our ability to generalize beyond training data to novel, out-of-distribution, image degradations is a hallmark of primate vision. The predictive brain, exemplified by predictive coding networks (PCNs), has become a prominent neuroscience theory of neural computation. Motivated by the recent successes of variational autoencoders (VAEs) in machine learning, we rigorously derive a correspondence between PCNs and VAEs. This motivates us to consider iterative extensions of VAEs (iVAEs) as plausible variational extensions of the PCNs. We further demonstrate that iVAEs generalize to distributional shifts significantly better than both PCNs and VAEs. In addition, we propose a novel measure of recognizability for individual samples which can be tested against human psychophysical data. Overall, we hope this work will spur interest in iVAEs as a promising new direction for modeling in neuroscience.

## 1 Introduction

Deep feedforward neural networks have profoundly impacted the development of computational neuroscience models of vision and have become de facto models of core object recognition. Despite these successes, it is also becoming increasingly clear that current deep neural networks remain outmatched by the primate brain's power and versatility [35]. The gap between human and machine vision is particularly obvious when artificial vision systems are required to generalize beyond training data to novel conditions including novel object transformations, occlusion, 3D viewpoints, or other image degradations not seen during training [9, 12, 38, 1].

What brain mechanisms allow primate vision to generalize beyond training distributions to novel, out-of-distribution, image degradations? An increasingly large body of cognitive neuroscience literature points to a critical role for cortical feedback as a key mechanism to help solve difficult recognition problems [30, 43, 36, 17, 23]. However, the computational principles underlying feedback mechanisms are not well understood. Starting with Helmholtz's unconscious inference, predictive processing has now become one of the most prominent theories of neural computation (see [16, 39] for reviews). The theory has taken multiple instantiations [32, 45, 11] but one of the core ideas is that the visual system learns generative models of the world – casting vision as an active inference process. Predictive coding networks (PCNs) have now become popular computer vision algorithms because of their underlying biological basis [2] and ability to explain behavioral data [27, 41, 5]. In these models, feedback connections aim to reconstruct lower level (bottom-up) visual representations based on abstract (top-down) a priori knowledge derived from a brain's internal model. This mechanism is thought to bring out-of-distribution generalization by actively leveraging top-down knowledge to correct for distributional shifts that arise with novel image degradations [4, 6].

2nd Workshop on Shared Visual Representations in Human and Machine Intelligence (SVRHM), NeurIPS 2020.

In parallel, progress in machine learning in the area of deep generative modeling has also been significant. In particular, variational autoencoders (VAEs) leverage amortized inference to model complex data [19] and have witnessed a widespread use in computer vision due to their high scalability [13, 40]. Recent refinements include iterative extensions of VAEs (iVAEs) which leverage stochastic variational inference (SVI) and achieve greater performance in unsupervised learning [24, 18, 29]. While predictive coding and variational autoencoders appear to share some superficial resemblance [28], a formal connection between the two has not been made explicit.

In this paper, we formally establish a connection between predictive coding and modern variational inference algorithms and consider the plausibility of iVAEs as computational neuroscience model extensions of PCNs. Furthermore, we establish the superiority of iterative models, iVAEs and PCNs, over classic VAEs for out-of-distribution generalization. We posit that the iterative and sample-specific inference of the iVAEs and PCNs is a key mechanism to gradually correct for the distributional shifts observed with out-of-distribution samples. We experimentally validate this hypothesis and propose a new quantitative measure of image recognizability derived from the number of inference steps required for iVAEs to generate a recognizable reconstruction. We hope to test these predictions in future psychophysics experiments and to motivate further discussions on the viability of iVAE as a biologically plausible vision model.

## 2 From PCN to iVAE

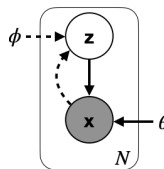

Here, we formalize mathematically the link between PCN and iVAE – both in terms of their loss functions and their learning mechanisms. We consider a Bayesian network with 2 random variables as shown in Fig. 1. The following theoretical derivation could be trivially extended to deeper hierarchical Bayesian networks. Let us consider a dataset $\boldsymbol{X} = \{\boldsymbol{x}^{(i)}\}_{i=1}^{N}$ composed of $N$ i.i.d samples of a random variable $\boldsymbol{x}$. We assume that $\boldsymbol{x}$ is generated by some random process involving an unobserved random variable $\boldsymbol{z}$ (see Eq. 1). The latent variable $\boldsymbol{z}$ is sampled from a Gaussian distribution (see Eq. 2). The mean of the likelihood is parametrized by $\boldsymbol{\mu}_\theta$ (in which $\theta$ denotes the parameters) and its variance is considered constant.

Figure 1: **Directed graphical model under consideration.** Solid lines denote the generative model, dashed lines denote the variational approximation of the true posterior. $\phi$ and $\boldsymbol{\theta}$ are the set of variational parameters, amortized over the $N$ training samples.

$$\boldsymbol{x} \sim p_\theta(\boldsymbol{x} \mid \boldsymbol{z}) \quad \text{s.t} \quad p_\theta(\boldsymbol{x} \mid \boldsymbol{z}) = \mathcal{N}\big(\boldsymbol{x}; \boldsymbol{\mu}_\theta(\boldsymbol{z}), \boldsymbol{\sigma}_x^2\big) \quad (1)$$

$$\boldsymbol{z} \sim p(\boldsymbol{z}) \quad \text{s.t} \quad p(\boldsymbol{z}) = \mathcal{N}\big(\boldsymbol{z}; \boldsymbol{\mu}_p, \boldsymbol{\sigma}_p^2\big) \quad (2)$$

**The PCN loss is a particular case of the ELBO.** Variational inference algorithms such as VAE approximate the true posterior with a family of distributions $q_\phi(\boldsymbol{z} \mid \boldsymbol{x})$ parameterized by $\phi$ that match the latent variable density distribution over the entire training set [21]. Those algorithms maximize the log-likelihood of the marginalized input probability distribution by maximizing the evidence lower bound (ELBO). The ELBO is commonly formalized as (see Demonstration S1):

$$ELBO(\boldsymbol{x}, \theta, \phi) = \mathbb{E}_{q_\phi(\boldsymbol{z}|\boldsymbol{x})}[\log p_\theta(\boldsymbol{x} \mid \boldsymbol{z})] - \text{KL}\big(q_\phi(\boldsymbol{z} \mid \boldsymbol{x}) \| p(\boldsymbol{z})\big) \quad (3)$$

As shown in Demonstration S2, the ELBO can be re-written as:

$$ELBO(\boldsymbol{x}, \theta, \phi) = \int_{\boldsymbol{z}} q_\phi(\boldsymbol{z} \mid \boldsymbol{x}) \log p(\boldsymbol{x}, \boldsymbol{z}) dz - \int_{\boldsymbol{z}} q_\phi(\boldsymbol{z} \mid \boldsymbol{x}) \log q_\phi(\boldsymbol{z} \mid \boldsymbol{x}) dz \quad (4)$$

Below we show that PCN constitutes a specific choice of the posterior estimate such that $q_\phi(\boldsymbol{z} \mid \boldsymbol{x})$ is a delta distribution centered at the most likely latent variable $\boldsymbol{z}^*$ [10, 3]:

$$q_\phi(\boldsymbol{z} \mid \boldsymbol{x}) = \delta_{\boldsymbol{z}^*}(\boldsymbol{z}) \text{ s.t. } \boldsymbol{z}^* = \underset{z}{\text{argmax}}(p(\boldsymbol{z} \mid \boldsymbol{x})) \text{ and } \delta_{\boldsymbol{z}^*}(\boldsymbol{z}) = \begin{cases} +\infty & \text{if } \boldsymbol{z} = \boldsymbol{z}^* \\ 0 & \text{else} \end{cases} \quad (5)$$

We then derive the PCN loss by replacing the approximate posterior by the delta distribution in Eq. 5:

$$ELBO(\boldsymbol{x}, \theta, \boldsymbol{z}^*) = \int_{\boldsymbol{z}} \delta_{\boldsymbol{z}^*}(\boldsymbol{z}) \log p(\boldsymbol{x}, \boldsymbol{z}) dz - \int_{\boldsymbol{z}} \delta_{\boldsymbol{z}^*}(\boldsymbol{z}) \log \delta_{\boldsymbol{z}^*}(\boldsymbol{z}) dz \tag{6}$$

$$= \log p(\boldsymbol{x}, \boldsymbol{z}^*) + C_1$$

$$= \log p_\theta(\boldsymbol{x} \mid \boldsymbol{z}^*) + \log p(\boldsymbol{z}^*)$$

$$= -\frac{1}{2\sigma_x^2} \big\| \boldsymbol{x} - \mu_\theta(\boldsymbol{z}^*) \big\|^2 - \frac{1}{2\sigma_p^2} \big\| \boldsymbol{z}^* - \mu_p \big\|^2 - \log|\sigma_x| - \log|\sigma_p| + C_2 \tag{7}$$

where $C_1$ and $C_2$ are constants. Note that the second integral on the right-hand side of Eq. 6 is constant as the output of a Dirac function is independent of its center, consequently it has no impact over the minimization process of the ELBO. The PCN loss as defined originally by Rao & Ballard [32] is shown in Eq. 8:

$$\mathcal{L}_{PC}(\boldsymbol{x}, \theta, \boldsymbol{z}^*) = \frac{1}{2\sigma_x^2} \big\| \boldsymbol{x} - \mu_\theta(\boldsymbol{z}^*) \big\|^2 + \frac{1}{2\sigma_p^2} \big\| \boldsymbol{z}^* - \mu_p \big\|^2 \tag{8}$$

Under the same hypothesis as the PCN, stating that the variance of both the likelihood and the prior are constant (and equal to unit variance), the maximization of Eq. 7 becomes equivalent to the minimization of Eq. 8. The link between PCN and VAE becomes evident: the PCN loss is a special case of the VAE loss (i.e., the negative ELBO) with PCN using a point-wise estimate instead of an approximate posterior distribution. Henceforth, we define the VAE loss using $\beta$ as the disentanglement factor [14] in Eq. 9 ($\beta$=1 by default).

$$\mathcal{L}_{VAE}(\boldsymbol{x}, \theta, \boldsymbol{\psi}) = -\mathbb{E}_{q(\boldsymbol{z};\boldsymbol{\psi})}[\log(p_\theta(\boldsymbol{x} \mid \boldsymbol{z}))] + \beta \, \mathrm{KL}\big(q(\boldsymbol{z};\boldsymbol{\psi}) \| p(\boldsymbol{z})\big) \text{ s.t } \boldsymbol{\psi} = \big(\mu_\phi(\boldsymbol{x}), \sigma_\phi^2(\boldsymbol{x})\big) \tag{9}$$

**Learning and inference in PCN, VAE and iVAE.** In PCN, the minimization of Eq. 8 is performed using the Expectation-Maximization (E-M) scheme [10]. The E-step, corresponding to inference, leads to an estimate of the most probable hypothesis $\boldsymbol{z}^*$ given the input $\boldsymbol{x}$ using $K$ gradient descent steps w.r.t. to the latent variable. The M-step, which corresponds to learning, uses one step of gradient descent w.r.t. $\theta$ to search for the model parameters that minimize the objective over the entire training set (see Alg. 1 for more details).

VAE, as an amortized variational inference model, learns the parameters of the posterior and the likelihood that are common to the entire training set [21, 33]. These amortized parameters are known to lead to a looser lower bound of the input distribution. This phenomenon is called the amortization gap [7]. Nevertheless, these methods provide highly scalable frameworks and fast inference (see Alg. S3 for more details).

Recently, the iterative VAE (iVAE) was shown to mitigate the amortization gap [24, 18, 29]. In the iVAE inference scheme, the amortized posterior parametrization serves as an initialization for the stochastic variational inference (SVI). Interestingly, the SVI is similar to the E-step used in the PCN: both estimate instance-specific latent variables iteratively. Said differently, the SVI could be considered as a variational version of the E-M algorithm [15].

| **Algorithm 1: PCN** | **Algorithm 2: iVAE** |
|---|---|
| **Input:** sample $\boldsymbol{x}$ of dataset $\boldsymbol{X}$, model parameters $\theta$, learning rates $\eta_z$ and $\eta$, inference steps $K$ 
 Initialize: $\boldsymbol{z}_0 = \boldsymbol{0}$ 
 **for** k = 1 .. K **do** 
 $\quad\mid \boldsymbol{z}_{k+1} = \boldsymbol{z}_k - \eta_z \nabla_{\boldsymbol{z}} \mathcal{L}_{PC}(\boldsymbol{x}, \theta, \boldsymbol{z}_k)$ 
 $\theta = \theta - \eta \nabla_\theta \mathcal{L}_{PC}(\boldsymbol{x}, \theta, \boldsymbol{z}_K)$ | **Input:** sample $\boldsymbol{x}$ of dataset $\boldsymbol{X}$, model parameters $\theta, \phi$, learning rates $\eta_\psi$ and $\eta$, inference steps $K$ 
 Initialize: $\boldsymbol{\psi}_0 = \big(\mu_\phi(\boldsymbol{x}), \sigma_\phi^2(\boldsymbol{x})\big)$ 
 **for** k = 1 .. K **do** 
 $\quad\mid \boldsymbol{\psi}_{k+1} = \boldsymbol{\psi}_k - \eta_\psi \nabla_{\boldsymbol{\psi}} \mathcal{L}_{VAE}(\boldsymbol{x}, \theta, \boldsymbol{\psi}_k)$ 
 $\theta = \theta - \eta \nabla_\theta \mathcal{L}_{VAE}(\boldsymbol{x}, \theta, \boldsymbol{\psi}_K)$ 
 $\phi = \phi - \eta \nabla_\phi \mathcal{L}_{VAE}(\boldsymbol{x}, \theta, \boldsymbol{\psi}_0)$ |

The link between PCN, SVI and iVAE becomes straightforward: they all share an iterative and sample-specific mechanism to infer an approximate posterior. In contrast, VAE learns a posterior distribution over the entire training set. The SVI can be seen as a variational extension of the PCN. In addition to this SVI process, the iVAE includes an additional amortized initialization step of the posterior.

# 3   Out-of-distribution generalization: definition and intuition

In out-of-distribution generalization tasks, a model must, at test time, generalize to new data distributions that were not encountered during training. Out-of-distribution generalization can be cast as an invariance problem: similar training and testing samples should both elicit the same representation even if they are drawn from different distributions [22]. In this paper, we simplify the out-of-distribution generalization task by only considering the robustness to relatively simple distributional changes such as additive noise, Gaussian blurring or salt & pepper degradation.

We posit that the amortized parameters of the likelihood term, which reflect the training likelihood distribution, give to the ELBO optimization surface a high degree of invariance w.r.t the input perturbation. To test this hypothesis, we first train VAE and iVAE toy models (with 2D latent variables models only) on the original (non-degraded) MNIST dataset. We then evaluate the optimization surface of both models with degraded inputs (see Fig. 2). Importantly, during evaluation, none of the models are given access to the non-degraded input. Fig. 2 shows that the topology of the optimization surface (i.e. the ELBO) is relatively invariant to the input degradation.

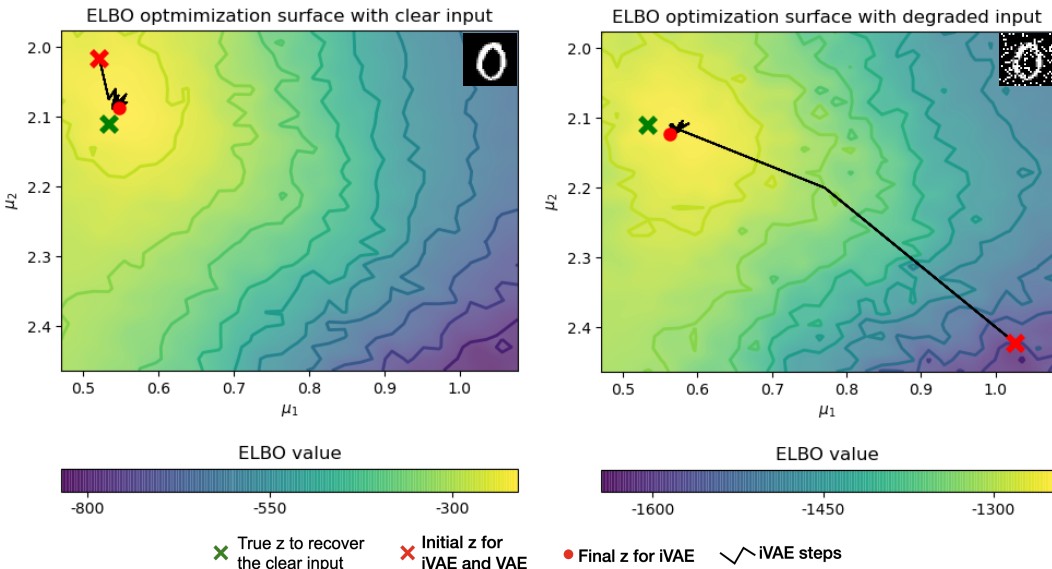

Figure 2: **ELBO optimization surfaces**. We train a VAE and iVAE with two latent variables on non-degraded inputs. The ELBO landscape was then computed by evaluating the function $\mathcal{L}_{VAE}(x, \theta, (\mu_1, \mu_2))$, in which $x$ is a non-degraded input for the map on the left, and a degraded input (with salt & pepper noise) for the map on the right.

Intuitively, since the parameters of the VAE are specific to the training distribution, the generated posteriors for out-of-distribution samples won't be aligned with the likelihood distribution as this is the case for training samples. Therefore, in such a situation, the reconstruction generated by the VAE is likely to be shifted away from the training distribution (see red cross in the left map of Fig. 2 away from green cross). In contrast, the iVAE (SVI and PCN) have the ability to move in the ELBO landscape towards posteriors that are maximizing the likelihood probability. We postulate that this hypothesis refinement mechanism should lead to a better out-of-distribution generalization capability. The following section is an empirical verification of this hypothesis.

# 4   Experiments

**Methods.**   As a baseline, we also report the classification accuracy for corrupted images (this baseline is denoted CL in Fig. 3-b). The SVI, PCN, VAE and iVAE were trained on clean images from the MNIST dataset [26] (see S4 for details regarding the choice of architectures and hyperparameters). We subsequently evaluated the models with MNIST test images that we corrupted

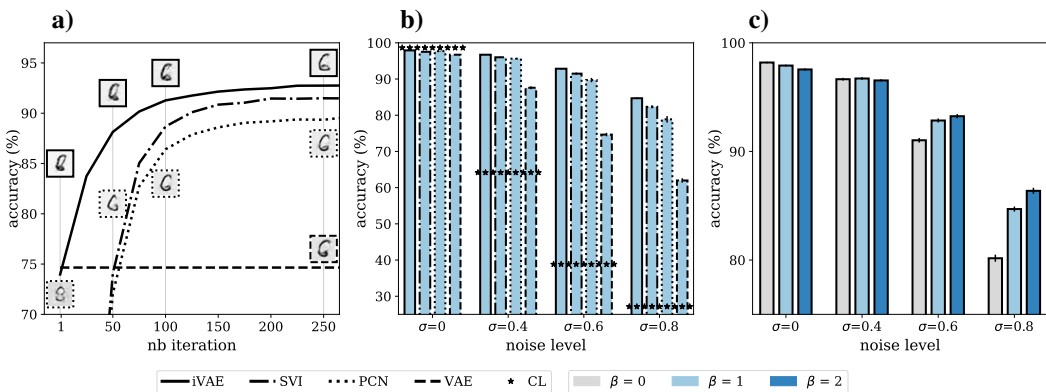

Figure 3: **Classification accuracy of the out-of-distribution sample reconstructions. a)** Accuracy over inference steps of images corrupted with white noise ($\sigma$=0.6). **b)** Accuracy for different noise levels. **c)** Impact of the disentanglement factor $\beta$ on accuracy under different white noise levels.

with varying levels of noise. Three types of noise degradations were used: Gaussian additive noise (white noise), salt & pepper noise (pixel intensities sampled from a Bernoulli distribution) and Gaussian blurring (convolution with a Gaussian kernel). Note that during evaluation, none of the models are given access to the original (and non-degraded) training distribution. Out-of-distribution generalization was evaluated as the accuracy of a convolutional neural network (CNN) fed with image reconstructions from the three generative models. Importantly, this CNN classifier was only trained on the original MNIST dataset so as to properly assess the generative models' abilities to preserve the classifier decision boundary. Other metrics could be used to evaluate out-of-distribution generalization (e.g. the $\ell_2$-norm with the non-degraded input), we choose the classification accuracy metric as it leads to a convenient parallel with behavioral data that can be collected from human participants during a psychophysics experiment (see subsection '*iVAE provides a testable prediction of the predictive brain theory)*'.

**iVAE outperforms SVI, PCN and VAE in out-of-distribution generalization.** Fig. 3-a shows that the accuracy of SVI, PCN and iVAE increases over inference steps and exceeds that of VAE by a large margin. The perceptual quality of the models' reconstructions also improves with more inference steps (see Fig. S6 for additional examples). The accuracies of iVAE's initial inference step and VAE's single inference step are relatively similar due to the amortized initialization of the iVAE. In contrast, the PCN's and SVI's initial accuracy are at the chance level because of the random initialization of the latent variable. At the final inference step, we observe that SVI constantly outperforms PCN in terms of accuracy. This suggests that the variational estimation of the posterior in SVI is more efficient than the point estimate used in PCN – at least in terms of out-of-distribution generalization. Since iVAE combines an amortized initialization with a variational iterative inference process, it out-performs systematically PCN, SVI and VAE as shown in Fig. 3-b (see Fig. S7 for the accuracy of the models on different types of noise). Furthermore, the accuracies of all generative models largely exceed the classifier baseline on noisy images. This gap is attributed to the models' abilities to push out-of-distribution latent variables back in the original distribution (see Fig. 2 for an illustration). We also observe that higher noise levels increase the performance gap between VAE and the iterative models (i.e., iVAE, SVI and PCN). This suggests that such iterative inference processes are crucial for out-of-distribution generalization. In Fig. 3-c, the disentanglement factor $\beta$ which controls the prior constraint is varied under different noise levels. As one might expect, the resulting accuracy is positively influenced by the prior when the noise level is high, and negatively when it is low. All reported results are consistent across noise types (see Fig. S8) with the exception of the Gaussian blurring where increasing $\beta$ has no significant effect on the classification accuracy.

**iVAE needs more inference iterations to classify atypical samples.** The ELBO measures the likelihood of an input image belonging to the training distribution. Therefore, the ELBO should be able to capture the level of typicality of individual samples with more prototypical (resp. atypical) samples associated to higher (resp. lower) ELBO values. Fig. 4-a qualitatively illustrates that higher

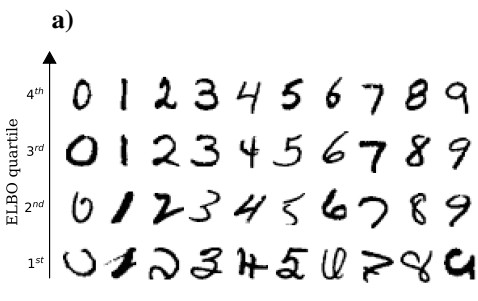 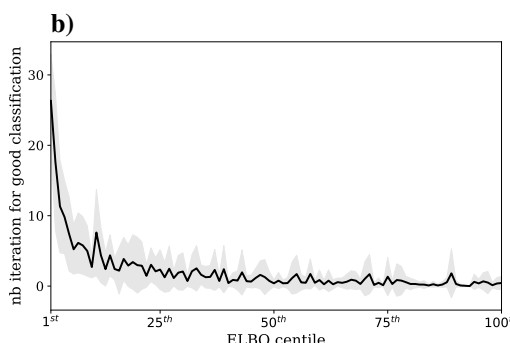

Figure 4: **Linking ELBO, sample typicality, and number of inference steps needed for a correct classification. a)** Samples of each digit class extracted from different quartiles of the ELBO. **b)** Number of inference steps needed before the model's reconstruction is correctly classified as a function of the ELBO centiles. The shaded area denotes the standard deviation across different digit classes.

ELBO digits appear more prototypical while lower ELBO digits appear more atypical or ambiguous. Interestingly, we also find empirically that iVAE allows us to draw a link between the ELBO and the number of inference steps before the output reconstruction is correctly classified (see Fig. 4-b).

**iVAE provides a testable prediction of the predictive brain theory.** Psychophysical data from prior studies have shown that observers' response times increases exponentially as the visibility of a stimulus is being reduced either by reducing the luminance [31] and/or the contrast [8]. Our results suggest that the model response time measured by the number of iterations needed to achieve good classification accuracy follows a similar trend when expressed as a function of the ELBO (see Fig. 4-b). In addition, the ELBO appears qualitatively to also provide a good measure of a stimulus prototypicality (see Fig. 4-a). While recognizability in psychophysics is often modeled by the distance to a classifier decision boundary, exemplars or prototypes [34], we propose the ELBO as an alternative measure of recognizability. This is consistent with a recent psychophysical study [37] which suggests that generative models account well for the perception of surface glossiness.

The backward masking protocol (i.e., reducing the visibility of a stimulus by presenting a noise mask shortly after the stimulus onset) is commonly used in psychophysics experiments to alter the effect of the feedback connections [25]. Masking has been shown to strongly impair recognition under challenging conditions including object occlusion and blurring [42] and it is widely believed that feedback mechanisms are necessary to help disambiguate degraded stimuli [30, 43, 36, 17, 23]. This is consistent with the results shown for the iVAE in Fig. 3-c.

# 5 Conclusion

We have shown that the iVAE constitutes a variational extension of the PCN. In addition, we show that the iVAE significantly outperforms SVI, PCN and VAE in out-of-distribution generalization. Finally, we show that the ELBO of the iVAE could serve as a novel measure of recognizability for individual samples which can be tested against human psychophysical data. Nevertheless, the link between the ELBO and standard cognitive science models based on exemplars and prototypes is still an open question [34].

As a next step, we believe that a hierarchical iVAE would provide a better model of primate vision given the hierarchical organization of our own visual system; it is also likely to be necessary to extend this work to natural images. Such an extension would allow to learn prior distributions directly from data and thus should provide more accurate predictions on the role of the feedback connection in the brain. Last, but not least, we have suggested how specific measures of recognizability could be derived from the iVAE model resulting in testable predictions for psychophysics which we plan to test in future work. Overall, we hope to spur interest from the community in considering iVAEs as a promising new direction for modeling vision beyond the feedforward sweep.

## Broader impact statement

A key question for computational neuroscience is to understand the role of cortical feedback which is currently poorly understood. By making explicit connections between cortical feedback and deep generative models, the present work may help further our understanding of brain mechanisms.

## Acknowledgment

This work was funded by ANITI (Artificial and Natural Intelligence Toulouse Institute, ANR-19-PI3A-0004).

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

## Supplementary Information

### S1 : Derivation of the ELBO

The objective of VAE is to estimate the true posterior $p(\boldsymbol{z} \mid \boldsymbol{x})$ with $q_\phi(\boldsymbol{z} \mid \boldsymbol{x})$ using the amortized parameters $\phi$. Using the Kullback-Leibler (KL) divergence between the two distributions, one can derive the evidence lower bound (ELBO) commonly used to optimize the variational parameters of a VAE:

$$
\begin{aligned}
\text{KL}\left(q_\phi(\boldsymbol{z} \mid \boldsymbol{x}) \| p(\boldsymbol{z} \mid \boldsymbol{x})\right) &= \int_{\boldsymbol{z}} q_\phi(\boldsymbol{z} \mid \boldsymbol{x}) \log \frac{q_\phi(\boldsymbol{z} \mid \boldsymbol{x})}{p(\boldsymbol{z} \mid \boldsymbol{x})} dz \\
&= \int_{\boldsymbol{z}} q_\phi(\boldsymbol{z} \mid \boldsymbol{x}) \log \frac{q_\phi(\boldsymbol{z} \mid \boldsymbol{x}) p(\boldsymbol{x})}{p(\boldsymbol{x}, \boldsymbol{z})} dz \\
&= \int_{\boldsymbol{z}} q_\phi(\boldsymbol{z} \mid \boldsymbol{x}) \log \frac{q_\phi(\boldsymbol{z} \mid \boldsymbol{x}) p(\boldsymbol{x})}{p_\theta(\boldsymbol{x} \mid \boldsymbol{z}) p(\boldsymbol{z})} dz \\
&= -\int_{\boldsymbol{z}} q_\phi(\boldsymbol{z} \mid \boldsymbol{x}) \log p_\theta(\boldsymbol{x} \mid \boldsymbol{z}) dz + \int_{\boldsymbol{z}} q_\phi(\boldsymbol{z} \mid \boldsymbol{x}) \log \frac{q_\phi(\boldsymbol{z} \mid \boldsymbol{x})}{p(\boldsymbol{z})} dz \\
&\quad + \log p(\boldsymbol{x}) \\
&= -\mathbb{E}_{q_\phi(\boldsymbol{z} \mid \boldsymbol{x})}[\log(p_\theta(\boldsymbol{x} \mid \boldsymbol{z}))] + \text{KL}\left(q_\phi(\boldsymbol{z} \mid \boldsymbol{x}) \| p(\boldsymbol{z})\right) + \log p(\boldsymbol{x}) \\
&= -ELBO + \log p(\boldsymbol{x})
\end{aligned}
$$

By definition, the KL being positive, one can derive a lower bound on the marginal input distribution:

$$
\text{KL}\left(q_\phi(\boldsymbol{z} \mid \boldsymbol{x}) \| p(\boldsymbol{z} \mid \boldsymbol{x})\right) \geqslant 0 \implies \log p(\boldsymbol{x}) \geqslant ELBO
$$

### S2 : From Eq. 4 to Eq. 3

The ELBO formulation in Eq. 4 is less commonly used compared to the one in Eq. 3 but the two are strictly equivalent:

$$
\begin{aligned}
ELBO &= \mathbb{E}_{q_\phi(\boldsymbol{z} \mid \boldsymbol{x})}[\log p_\theta(\boldsymbol{x} \mid \boldsymbol{z})] - \text{KL}\left(q_\phi(\boldsymbol{z} \mid \boldsymbol{x}) \| p(\boldsymbol{z})\right) \\
&= \int_{\boldsymbol{z}} q_\phi(\boldsymbol{z} \mid \boldsymbol{x}) \log p_\theta(\boldsymbol{x} \mid \boldsymbol{z}) dz - \int_{\boldsymbol{z}} q_\phi(\boldsymbol{z} \mid \boldsymbol{x}) \log \frac{q_\phi(\boldsymbol{z} \mid \boldsymbol{x})}{p(\boldsymbol{z})} dz \\
&= \int_{\boldsymbol{z}} q_\phi(\boldsymbol{z} \mid \boldsymbol{x}) \log \frac{p_\theta(\boldsymbol{x} \mid \boldsymbol{z}) p(\boldsymbol{z})}{q_\phi(\boldsymbol{z} \mid \boldsymbol{x})} dz \\
&= \int_{\boldsymbol{z}} q_\phi(\boldsymbol{z} \mid \boldsymbol{x}) \log \frac{p(\boldsymbol{x}, \boldsymbol{z})}{q_\phi(\boldsymbol{z} \mid \boldsymbol{x})} dz \\
&= \int_{\boldsymbol{z}} q_\phi(\boldsymbol{z} \mid \boldsymbol{x}) \log p(\boldsymbol{x}, \boldsymbol{z}) dz - \int_{\boldsymbol{z}} q_\phi(\boldsymbol{z} \mid \boldsymbol{x}) \log q_\phi(\boldsymbol{z} \mid \boldsymbol{x}) dz \qquad (10)
\end{aligned}
$$

### S3 : VAE pseudo-code

---
**Algorithm 3:** VAE

---
**Input:** sample $\boldsymbol{x}$ of dataset $\boldsymbol{X}$, model parameters $(\theta, \phi)$, learning rate $\eta$, inference steps $K$
$\boldsymbol{\psi} = (\mu_\phi(\boldsymbol{x}), \sigma_\phi^2(\boldsymbol{x}))$
$\theta = \theta - \eta \nabla_\theta \mathcal{L}_{VAE}(\boldsymbol{x}, \theta, \boldsymbol{\psi})$
$\phi = \phi - \eta \nabla_\phi \mathcal{L}_{VAE}(\boldsymbol{x}, \theta, \boldsymbol{\psi})$

---

### S4 : Learning and model parameters

PCN, VAE and iVAE leverage the standard generative model settings of VAE as described in the following equations:

$$
p_\theta(\boldsymbol{x} \mid \boldsymbol{z}) = \mathcal{N}\left(\boldsymbol{x}; \boldsymbol{\mu}_\theta(\boldsymbol{z}), \boldsymbol{I}\right) \qquad (11)
$$

$$
p(\boldsymbol{z}) = \mathcal{N}\left(\boldsymbol{z}; \boldsymbol{0}, \boldsymbol{I}\right) \qquad (12)
$$

All encoder (i.e. $\boldsymbol{\psi}$) and decoder (i.e. $\boldsymbol{\mu}_\theta$) models use three fully connected layers with tanh activations, hidden dimensions $h_1 = 512$, $h_2 = 256$ and latent dimension $z = 15$. For the VAE and iVAE, $\beta$ was set by default to 1 and when specified we vary $\beta$ with the following values : $[0.0, 1.0, 2.0]$. SVI in the iVAE is run for 20 iterations with a inference update rate $\eta_\psi = 10^{-2}$ during training and 500 iterations with a inference update rate of $\eta_\psi = 10^{-3}$ during evaluation. For the PCN, the total number of inner-loop iterations was set to 100 during training and 500 during evaluation with $\eta_z = 10^{-2}$. We used the MNIST dataset with normalized pixel values. Fig. 4-b was produced with the $60,000$ test samples of the QMNIST dataset [44]. All models were trained for 200 epochs with a batch size of 1,024, a learning rate $\eta = 10^{-3}$. For all models, we use Adam [20] as an optimizer both for inference and learning. The classifier's architecture and training parameters were the default used in the pytorch MNIST example.

**S5 : Noise parameters.** Examples of an MNIST digit corrupted with different noise types and levels. The level of noise applied to an image is controlled with the standard deviation (denoted $\sigma$) of the Gaussian distribution for the white noise and the Gaussian blurring, and the probability of pixel's corruption (denoted $p$) for the salt & pepper degradation.

| Noise type | Parameters | | | |
|---|---|---|---|---|
| Noise level | 1 | 2 | 3 | 4 |
| Gaussian Blurring | $\sigma$=1 | $\sigma$=2 | $\sigma$=3 | $\sigma$=4 |
| White Noise | $\sigma$=0.2 | $\sigma$=0.4 | $\sigma$=0.6 | $\sigma$=0.8 |
| Salt & pepper | $p$=0.1 | $p$=0.2 | $p$=0.3 | $p$=0.4 |

**S6 : Examples of reconstructions.** Image reconstructions for different types of noise, and different number of iterations. The input is corrupted with blurring ($\sigma = 2$), with white noise ($\sigma = 0.6$) and with salt & pepper noise ($p = 0.4$).

| | Gaussian Blurring | | | White Noise | | | Salt & pepper | | |
|---|---|---|---|---|---|---|---|---|---|
| | iVAE | PCN | VAE | iVAE | PCN | VAE | iVAE | PCN | VAE |
| input | | | | | | | | | |
| t=1 | | | | | | | | | |
| t=250 | | | | | | | | | |
| t=500 | | | | | | | | | |
| target | | | | | | | | | |

**S7 : Classification accuracy of reconstructed images with different noise types and levels**. iVAE consistently outperforms PCN and VAE. "CL" denotes the classification accuracy for the unprocessed noisy images as a baseline. The equivalence between noise level and noise parameters for each type of noise is described in S5.

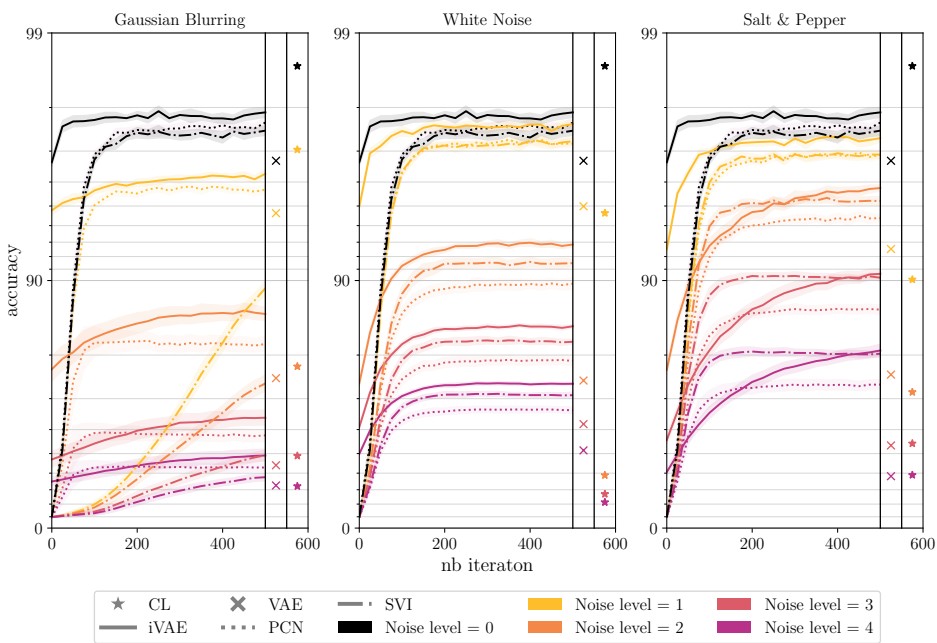

**S8 : Effect of the disentanglement factor of the classification accuracy.** The noise level use in the figure are the same than in S5 and S7. Note that increasing beta leads to better accuracy for global noise types (i.e. Gaussian noise and salt & pepper), and has no significant effect on local noise (Gaussian blurring)

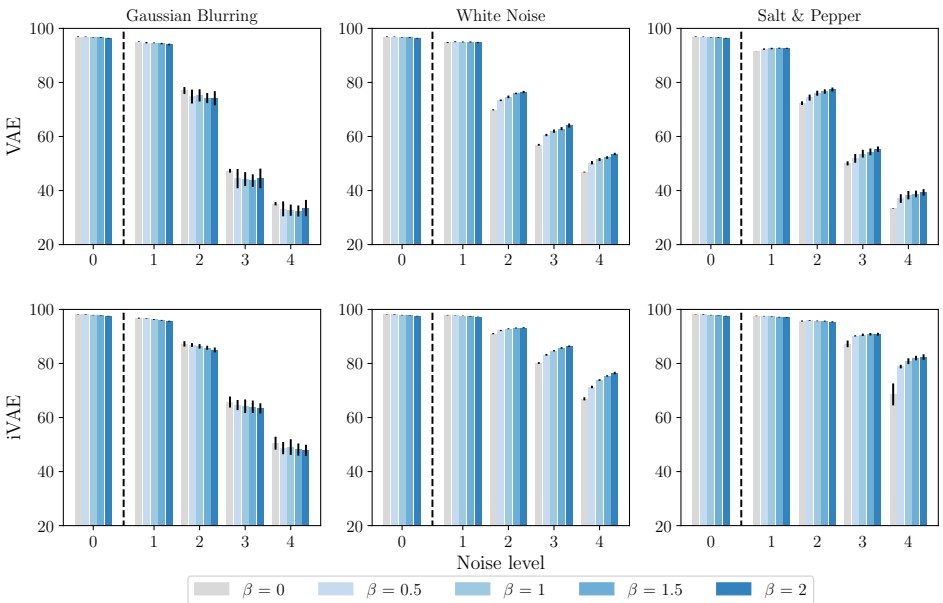

