# OpenReview forum: "Iterative VAE as a predictive brain model for out-of-distribution generalization"
_NeurIPS.cc/2020/Workshop/SVRHM — SVRHM@NeurIPS Poster_

### Official Review · AnonReviewer1 · 2020-10-27
**Interesting insight, novel model worth discussion**

**Rating:** 8
**Confidence:** 3

**Review:**

Interesting insight, novel model worth discussion

The paper draws a theoretical parallel between iterative VAE (iVAE) and predictive coding networks (PCN; a la Rao & Ballard 1999), thus linking iVAEs to a theory of neuronal coding. Further, the paper shows empirically that iVAE can recover noise-degraded MNIST digits for classification, improving upon both non-iterative VAE and PCN (in addition to the classifier itself without denoising). Thus, iVAE presents a theoretically and biologically-motivated model of recurrent processing and generalization in vision.

The paper is recommended for acceptance, as it offers an interesting perspective and viable model worth discussing in the workshop. It is not without limitations and questions (below), which do not take away from the interest of the paper for broader discussion.


Comments:
- The broad question concerns "generalization." Why is it defined in this specific way, as noise-removal and not, e.g., view-invariance? Also, why is the evaluation metric defined this way? Specifically:
- Why use a classifier to judge reconstruction? During training, the VAE/PCN models are unaware of classification (right?). Thus, to evaluate the models, why not use a more basic metric such as pixel-wise L2 loss, or another reconstruction loss used during training? Although the practical motivation may be to preserve classification accuracy, it is conterintuitive to evaluate models on something they're not trained on. To put the question another way, the finding seems surprising that these models preserve something so high-level; but this result could be entirely due to preserving much simpler features such as pixel-wise values. This simple explanation is not evaluated or accounted for.
- The empirical results seem to be not informative, but rather entirely expected from the theoretical results. Specifically, iVAE improves over iterations and outperforms VAE because 1) iVAE starts at similar performance to VAE; 2) iVAE has to improve during iteration because the objective guiding iterations is to improve the reconstruction. Along the same lines, is it surprising that iVAE outperforms PCN? The former has the advantage of an amortized initialization. Even if iVAE and PCN gain the same from iteration (which seems to be the case based on Fig 2a), iVAE will win by having a "headstart." To be sure, results being expectable does not take away from the theoretical interest, which is what enables prediction of the results in the first place. Also, seeing predicted results can be reassuring. But the significance of the empirical results should be put into perspective in text.
- More informative experiments can also be done. For example, there seems to be an opportunity to separately test the benefit of initialization and of using a full posterior, when comparing iVAE to PCN. Based on your nice theoretical results, iVAE can be reduced to PCN if only the MLE of the posterior is used. Thus, you could test an iVAE that is initialized normally but uses only the MLE during SVI.
- Part of the motivation of the paper is that "hierarchical iVAE would provide a better model of primate vision." If so, existing relevant results should be discussed, e.g., https://arxiv.org/pdf/1606.05579.pdf, https://www.biorxiv.org/content/10.1101/2020.06.16.155556v1.full.pdf, https://arxiv.org/pdf/2006.14304.pdf.


Minor comments:
- Typo in line 62: q_phi(x|z) should be q_phi(z|x).
- "Under the assumption that greater prior strength in the PCN and iVAE reflects the need for greater feedback [29]": the rationale for this assumption is not transparent, nor is it clear where it is mentioned in reference 29. Please specify.
- "how specific measures of recognizability could be derived from the iVAE model resulting in testable predictions for psychophysics which we plan to test in future work": Can you specify what testable prediction you plan to test? This statement is quite vague. In your own interpretations of the results, it is suggested that decision boundaries, prototypes, and ELBO are likely correlated. How can you actually distinguish them?
- "By making explicit connections between cortical feedback and deep generative models": The connection does not seem explicit, as it is unclear whether feedback relates to SVI (short time scale), to calculating the reconstruction error (both during SVI and during training), to both, or something else in your model. You also suggest that feedback may be related to the prior, but that is not explicit either, as commented above.

**Bio Award:**

Yes, paper should be nominated as I have given it a high score and it is also relevant to the award (presents a biologically-driven generative model).

---

### Official Review · AnonReviewer3 · 2020-10-28

**Rating:** 6
**Confidence:** 5

**Review:**

Summary and contributions: This paper derives the PCN objective from Rao & Ballard as a special case of the VAE objective assuming that (1) the approximating distribution q(z |) is a point estimate and (2) the variance of the likelihood and prior of the generative model are constant. The paper then observes that the EM algorithm used to train PCN and the iterative inference algorithms used to train iVAE follow the same algorithmic structure. The paper last presents experiments that show that iVAE outperforms PCN and VAE on out-of-distribution generalization.

Strengths:
1. presents a mathematical connection between the objective of PCN and VAE
2. notices that iVAE is simply implementing an unrolled algorithm like EM

Weaknesses:
1. while it is worth formalizing the connection between PCN and iVAE, the connection seems rather straightforward and not enormously novel, especially considering progress on recent work in developing iterative inference methods with neural networks: Neural Expectation Maximization [1] implements the EM procedure described in algorithm 1 of this paper, and its successor IODINE [2] implements the iVAE procedure described in algorithm 2 of this paper. Dynamic versions of both have also been proposed: Relational Neural Expectation Maximization [3], and its iVAE counterpart OP3 [4].
2. That iVAE would perform better in out-of-distribution generalization also seems like a straightforward claim, especially when we consider that iVAE is, in some sense, training on the testing set. However, I agree that it is valuable to have experiments that definitively show this.
3. I think I would have appreciated more context for how the connection between PCNs and VAEs motivate the authors' hypothesis that iVAE may be better for out-of-distribution generalization. The intro briefly mentioned that cortical feedback has been studied to be useful for solving difficult recognition problems, but it is not clear what the link is between PCNs and out-of-distribution generalization.

Recommendations:
1. as I see the theoretical component of this paper as rather straightforward, I belive this paper could have the most potential as a large-scale empirical study over multiple datasets on how iterative inference improves out-of-distribution generalization.
2. I think it is crucial to address Weakness 3 above; otherwise it is not clear how the theoretical portion of the paper connects to the empirical portion.

[1] Greff, K., Van Steenkiste, S., & Schmidhuber, J. (2017). Neural expectation maximization. In Advances in Neural Information Processing Systems (pp. 6691-6701).
[2] Greff, K., Kaufman, R. L., Kabra, R., Watters, N., Burgess, C., Zoran, D., ... & Lerchner, A. (2019). Multi-object representation learning with iterative variational inference. arXiv preprint arXiv:1903.00450.
[3] Van Steenkiste, S., Chang, M., Greff, K., & Schmidhuber, J. (2018). Relational neural expectation maximization: Unsupervised discovery of objects and their interactions. arXiv preprint arXiv:1802.10353.
[4] Veerapaneni, R., Co-Reyes, J. D., Chang, M., Janner, M., Finn, C., Wu, J., ... & Levine, S. (2020, May). Entity abstraction in visual model-based reinforcement learning. In Conference on Robot Learning (pp. 1439-1456). PMLR.

---

### Official Review · AnonReviewer2 · 2020-10-30
**Interesting links to human studies of categorization**

**Rating:** 6
**Confidence:** 5

**Review:**

The paper is very interesting in the sense that it takes the idea of semi amortization in context of variational autoencoders (VAEs) and studies how it might suggest computational accounts of human beings in recognizing out of distribution samples, specifically in a tradeoff of inference time and accuracy on the harder examples.

I have an alternative explanation for why iVAEs work better than VAEs for the noisy example case, which is not necessarily around “making out of distribution latent variables in distribution”, which in my opinion does not sound technically correct. Instead here is an alternative hypothesis to explain the observations from the paper:

When we add noise to images and feed them through the inference network of a VAE one expects the model to be better at supporting classification of images, since VAEs learn a latent space with compression, that is, low I(X; Z). However, the inference network of such models might not generalize to the out of distribution case, in which case semi-amortized or iterative VAEs provide a much better estimate of the posterior q(z| x) in the noisy case since they actually solve the optimization problem of interest (as opposed to just doing a feedforward pass through the inference network). In any case, with accurate inference the VAE should support classification of the noisy examples. This above explanation also accounts for why higher beta does better in the noisy case as higher beta means more compression, or lower I(X; Z) learnt by the model [A].

One technical issue in the derivation of the correspondence of the PCNs and VAEs appears to be that the derivation currently only seems to hold in the discrete case (which is the case for which the delta function is written the way it is right now). In the continuous case, the delta function at z = z* would have a pdf of (tending to) \inf making log of that also \inf (in the second term of the derivation). Might be worth clarifying that this is only the case for discrete z (which is admittedly a less common or useful case).


[A]: Alemi, Alexander A., Ben Poole, Ian Fischer, Joshua V. Dillon, Rif A. Saurous, and Kevin Murphy. 2017. “Fixing a Broken ELBO.” arXiv [cs.LG]. arXiv. http://arxiv.org/abs/1711.00464.

---

### Decision · Program_Chairs · 2020-11-02

Accept (Poster)

---

> ### Public Comment · ~Victor_Boutin2 · 2020-11-28
> **General response to reviewers**
>
> We thank the reviewers for their insightful feedback. We believe we have addressed all the points they have raised. We believe the manuscript has greatly improved as a result. Below are answers to their comments.
>
> **Reviewers suggest that out-of-distribution generalization seems straightforward as iVAE is, in some sense, trained on the test set.** The iVAE is NOT trained on the test set! During the test phase, the iVAE only sees the degraded input, not the original image. This point is crucial, and we agree it was not clear enough in the core text of the article. We updated the text accordingly.
>
> **Reviewers suggest that the theoretical correspondence of the PCNs and VAEs holds only for discrete distribution.** Our derivation still holds in the continuous case. The integral shown in Eq. 4 is constant w.r.t. to z* as the output of the Dirac function does not depend on where its mass is concentrated. Consequently, this integral does not affect the optimization problem. We have updated our paper to better emphasis this point.
>
> **Reviewers believe that robustness comes from compression.** We agree with the reviewer that the compression of the latent allows, to some extent, a better robustness of the classification accuracy to unseen test distribution. We think that the compression is crucial for the ELBO landscape to be invariant to perturbations (see new Fig.2). We now show that the iterative inference procedure in the iVAE helps get the posterior estimated on a degraded input to get closer to the posterior estimated on non-degraded images... We believe that the combination of these two mechanisms -- compression of the latent space combined with the iterative inference procedure -- is crucial to push the degraded input distribution towards the non-degraded training distribution.
>
> **Reviewers wonder why we define out-of-distribution generalization as noise-removal.** We define generalization broadly as robustness to any change in the input distribution that has not been seen during training. In this article we focus on relatively simple distributional changes including additive noise, Gaussian blurring or salt & pepper degradation to establish a proof of concept. We plan to extend our study to more complex and deeper networks to tackle more challenging distributional shifts (e.g., changes in viewing position). We have updated our paper to better define out-of-distribution generalization.
>
> **Reviewers wonder why we used classification accuracy to evaluate out-of-distribution generalization, and not other metrics that seem more relevante (e.g. l2-norm).** Here, classification is used as a *psychophysics measure* for possible comparison with human decisions since we cannot characterize neural representations directly.
>
> **Reviewers suggest that  empirical results are not informative because they seem to be expected from the theoretical framework.** Our theoretical work only links PC and iVAEs. The out-of-distribution robustness of iVAEs is only a hypothesis that is tested empirically. We have added a subsection to further motivate this intuition.
>
> **Reviewers suggest to run more informative experiments to assess separatly the effect of the variational inference and amortized initialization.** We thank the reviewer for suggesting novel experiments. We have ran additional simulations to compare the PCN with its variational counterpart (i.e., the SVI). We observed that SVI systematically outperforms PCN, confirming that the variational estimation of the posterior is beneficial for out-of-distribution generalization tasks. In addition, as the SVI is variational but do not leverage amortized initialization, we were able to observe that the amortized initialization of the iVAE is also beneficial in terms of classification accuracy on degraded inputs.
>
> **Reviewers suggest that the connection between PCN and iVAE seems straightforward and not enormously novel.** To the best of our knowledge, this is first time that a theoretical equivalence (both in terms of loss function and learning scheme) between iVAE and PC is shown. The main novelty of the article in lying in the link between a ML framework (iVAE) and a computational neuroscience theory (the Predictive Coding).
>
> **Reviewers would have appreciated more context for how the connection between PCNs and VAEs motivate the authors' hypothesis that iVAE may be better for out-of-distribution generalization.** In PCN networks, feedback connections aim to reconstruct lower level (bottom-up) visual representations based on abstract (top-down) a priori knowledge derived from a brain's internal model. This mechanism is thought to bring out-of-distribution generalization by actively leveraging top-down knowledge to correct for distributional shifts that arise with novel image degradations. As iVAE is tightly linked to PCN, it is then a good candidate for out-of-distribution generalization tasks. We have updated our introduction to emphasis this point.